# Intimate partner violence victimisation in early adulthood: psychometric properties of a new measure and gender differences in the Avon Longitudinal Study of Parents and Children

Alexa R Yakubovich,[1] Jon Heron,[2] Gene Feder,[2] Abigail Fraser,[2] David K Humphreys[1,3]

¹Social Policy & Intervention, University of Oxford, Oxford, UK
²School of Social and Community Medicine, University of Bristol, Bristol, UK
³Green Templeton College, University of Oxford, Oxford, UK

**Correspondence to**
Ms Alexa R Yakubovich; alexa.yakubovich@spi.ox.ac.uk

## ABSTRACT

**Objectives** To evaluate the psychometric properties of a novel, brief measure of physical, psychological and sexual intimate partner violence (IPV) and estimate the overall prevalence of and gender differences in this violence.

**Design** Data are from the Avon Longitudinal Study of Parents and Children (ALSPAC), a birth-cohort study.

**Setting** Avon, UK.

**Participants** 2128 women and 1145 men who completed the questionnaire assessment at age 21.

**Outcome measures** Participants responded to eight items on physical, psychological and sexual IPV victimisation at age 21. Participants indicated whether the violence occurred before age 18 and/or after and led to any of eight negative impacts (eg, fear). We estimated the prevalence of IPV and tested for gender differences using $\chi^2$ or t-tests. We evaluated the IPV victimisation measure based on internal consistency (alpha coefficient), dimensionality (exploratory factor analysis) and convergent validity with negative impacts.

**Results** Overall, 37% of participants reported experiencing any IPV and 29% experienced any IPV after age 18. Women experienced more frequent IPV, more acts of IPV and more negative impacts than men (p<0.001 for all comparisons). The IPV measure showed high internal consistency ($\alpha$=0.95), strong evidence for unidimensionality and was highly correlated with negative impacts (r=0.579, p<0.001).

**Conclusions** The prevalence of IPV victimisation in the ALSPAC cohort was considerable for both women and men. The strong and consistent gender differences in the frequency and severity of IPV suggest clinically meaningful differences in experiences of this violence. The ALSPAC measure for IPV victimisation was valid and reliable, indicating its suitability for further aetiological investigations.

## INTRODUCTION

Intimate partner violence (IPV) is the most common violence perpetrated against women worldwide with severe consequences, including mortality, injury and mental health

### Strengths and limitations of this study

► This study is the first to evaluate a novel and relatively brief measure of physical, psychological and sexual intimate partner violence using data from a long-running, high-quality birth-cohort study in the UK.

► Timing of violence was measured which allowed us to compute both the lifetime and early adulthood prevalence of intimate partner violence.

► We used a robust analysis strategy to test for gender differences in intimate partner violence, which included analysing the impacts of this violence to determine the severity of clinical burdens among women and men.

► Details on specific incidents or perpetrators of intimate partner violence were not measured and the generalisability of study findings to the national population and other contexts should be investigated.

disorders.[1–3] The most recent estimates for the UK indicate that IPV, especially among women, should be a public health priority, with 23% of women and 11% of men reporting any physical, psychological or sexual IPV in their lifetime.[4 5] However, designing interventions for IPV requires accurately measuring and understanding its burden. Unlike many public health problems, official (eg, police or hospital-reported) data typically provide poor estimates since most people do not contact formal services after experiencing IPV.[6]

Although survey data on IPV tend to be viewed as more accurate, measurement quality varies widely. While single-term and vague items such as 'violence' are insufficient to measure the complexity of IPV, multi-item scales vary in content and length. The most commonly used measure is currently the Conflict Tactics Scale,[7–9] which measures

specific behaviours by a current or previous dating, cohabiting or marital partner. However, the Conflict Tactics Scale has been criticised for measuring IPV only within the context of conflicts or disagreements and not measuring the intent (eg, self-defence or harm) or impact of violence.[10 11] Other validated scales include the Composite Abuse Scale,[12] WHO multicountry survey,[13] Abusive Behavior Inventory,[14] Severity of Violence Against Women Scale,[15] Measure of Wife Abuse[16] and the Extended-Hurt/Insult/Threaten/Scream tool.[17] However, several of these do not measure psychological IPV (including controlling behaviour)[13 15 17] and most are relatively long (>30 items),[12 14–16] risking response burden in larger or repeated-measures surveys.

In recent years, in response to the criticisms and limitations of existing measures, short-form measures of physical, psychological and sexual IPV have been developed with emerging evidence of validity (eg, among Canadian women: the short-form Composite Abuse Scale[18]). This study is the first psychometric evaluation of a short-form measure for physical, psychological and sexual IPV developed in the UK, which uniquely also collected data on the impacts of this violence and sampled both women and men. We aimed to (1) evaluate the psychometric properties of this new instrument; (2) estimate the overall prevalence of IPV and its impacts and (3) test for gender differences in a UK-based birth-cohort study. This is essential to developing aetiological evidence for IPV against women, which, as demonstrated by a recent systematic review of prospective-longitudinal studies, is severely limited outside the USA.[19]

## METHOD

We used data from the Avon Longitudinal Study of Parents and Children (ALSPAC). The birth-cohort study has established trust among participants, who have been self-completing questionnaires since age 5 (now in early adulthood)—which is ideal for measuring IPV. The sampling frame included all pregnant women resident in one of three health districts in Avon, UK due between 1 April 1991 and 31 December 1992.[20 21] The initial number of pregnancies enrolled was 14 541. When participating children were approximately age 7, eligible cases not in the study were contacted, increasing the sample to 15 427 pregnancies, with 14 775 live births (76% of eligible live births)—these children are our target sample. A searchable data dictionary is available at http://www.bristol.ac.uk/alspac/researchers/our-data/.

### Measuring IPV

At age 21, 3458 participants completed the online questionnaire, of whom 3273 (2128 women, 1145 men) provided any data on IPV, making this our starting sample. The IPV measures described below (see table 1) were based on a previous National Society for the Prevention of Cruelty to Children (NSPCC) questionnaire used in a young population in Bristol,[22] with modified wording and

additional items based on the PROVIDE questionnaire.[23] The development group consisted of IPV researchers (Christine Barter, Marianne Hester, Eszter Szilassy and GF); the questionnaire was piloted for acceptability with the ALSPAC participant advisory group.

### Main instrument: IPV victimisation

Eight items measured physical, psychological and sexual IPV victimisation. A ninth item (feeling scared) was relevant to the impact of this violence and is therefore included with the impact items. Participants indicated the frequency of each item (0=never to 3=often) and whether the behaviour occurred before and/or after age 18, allowing for measurement of temporality.

### Impacts of IPV

Ten items measured the psychosocial impacts of IPV. Eight items indicated negative impacts (eg, upset). One item measured whether the violence had no effect and two measured positive effects (eg, feeling loved).

### Analysis

For aim 1, we evaluated the internal consistency, dimensionality and convergent validity of the IPV victimisation scale. To determine internal consistency, we computed an alpha coefficient for the eight IPV victimisation items using the polychoric (rather than Pearson) correlation matrix, which accounts for variables being ordinal rather than continuous.[24] As the scale's dimensionality was unknown, we conducted an exploratory factor analysis using this matrix.[25] We decided the appropriate number of factors based on their eigenvalues (using Kaiser's criterion that >1 indicates a viable factor), scree plot and theoretical plausibility.[26] If a two (or more) factor solution was favourable, we decided a priori to use oblique rotation since we expected differing dimensions of abuse to correlate. To test for possible gender differences in factor solutions, we also ran the exploratory factor analysis separately for women and men. To assess convergent validity, we computed the Pearson correlation between the average frequency of IPV experiences and sum total of negative impacts among those who had experienced any IPV. For this step, we first confirmed (via polychoric correlation) that the negative impacts of IPV were positively correlated with each other and negatively correlated with the positive and null impacts (table 1).

For aim 2, we computed the mean of participants' scores across the eight IPV items (reflecting the average frequency of IPV experiences, 0–3), the mean number of IPV acts experienced at least once (0–8), the mean number of negative impacts (0–8), the proportion of participants who experienced any IPV and the prevalence of any IPV with at least one negative impact. For aim 3, we tested for gender differences in each item and summary variable using $\chi^2$ or two-sided t-tests, as appropriate. For the latter, when the Levene's test suggested that the variances of women's and men's scores were unequal, we computed a two-sided t-test for unequal variances.

**Table 1** IPV victimisation and impact items

| Order | Victimisation items: How often altogether have any of your partners ever done any of the following to you and how old were you? | Type of IPV |
|---|---|---|
| 1 | Told you who you could see and where you could go and/or regularly checked what you were doing and where you were (by phone or text)? | Psychological |
| 2 | Made fun of you, called your hurtful names, shouted at you? | Psychological |
| 3 | Used physical force such as pushing, slapping, hitting or holding you down? | Physical |
| 4 | Used more severe physical force such as punching, strangling, beating you up, hitting you with an object? | Physical |
| 5 | Pressured you into kissing/touching/something else? | Sexual/psychological |
| 6 | Physically forced you into kissing/touching/something else? | Sexual |
| 7 | Pressured you into having sexual intercourse? | Sexual/psychological |
| 8 | Physically forced you into having sexual intercourse? | Sexual |

| Order | Impact items: How did you feel after they did these things to you? | Dimension |
|---|---|---|
| 1 | Did any of the above make you feel scared or frightened, or did any partner make you feel frightened in any other way?* | Negative |
| 2 | Upset/unhappy | Negative |
| 3 | Affected my work/studies | Negative |
| 4 | Made me feel sad | Negative |
| 5 | No effect/not bothered | Null |
| 6 | Anxious | Negative |
| 7 | Made me drink more alcohol/take more drugs | Negative |
| 8 | Felt loved/protected/wanted | Positive |
| 9 | Thought it was funny | Positive |
| 10 | Angry/annoyed | Negative |
| 11 | Depressed | Negative |

For each victimisation item, participants indicate the frequency of occurrence—where 0=never, 1=once, 2=a few times, 3=often—and age of occurrence, where 1=under 18, 2=over 18, 3=both. The question prompt included the following definition for partner: 'By partner we mean anyone you have ever been out with or had a relationship with, long term or short term (including one night stands)'. For each impact item, participants indicated 'yes' or 'no' as to whether this is how the IPV they experienced affected them.
*This item was asked along with the victimisation items and was therefore measured on the 'frequency' response scale.
IPV, intimate partner violence.

## Patient and public involvement

The IPV measure was based on the NSPCC questionnaire,[22] which was developed with a young person's advisory group, and the PROVIDE survey,[23] which was discussed with the PROVIDE patient and public involvement group. Additionally, ALSPAC has an advisory panel of >30 participants who meet bimonthly to advise on study design, methodology and acceptability. ALSPAC communicates with participants via regular newsletters and has an active website and social media presence.

## RESULTS

Table 2 summarises sample characteristics by gender. Women and men were very similar on baseline sociodemographics: most were white and had characteristics of higher socioeconomic status. At age 21, most women and men saw themselves as completely heterosexual (83% women, 85% men), followed by a smaller proportion reporting at least some same-sex preferences (16%

women, 13% men) and a small number indicating asexuality (<1%). More women (72%) than men (59%), however, had been in relationships longer than 3 months by age 18 and, by age 20, more women (12%) than men (6%) were living with partners or children.

## Reliability and validity

Correlations were strong between all IPV scale items, ranging from 0.57 (between experiencing humiliation/name-calling/shouting and forced sexual touch) to 0.92 (between forced and coerced touch) (online supplementary appendix table A1). The alpha coefficient was 0.95, indicating strong internal consistency.

The exploratory factor analysis suggested a one or two-factor solution (see online supplementary appendix table A2 for factor loadings). Only the first factor had an eigenvalue more than 1 (5.834). All items loaded highly onto this factor (ranging from 0.771 to 0.898), which suggests that using a factor-based score for experiences of IPV overall would be a valid analytical method in this

**Table 2** Sociodemographic characteristics of the sample by gender

| | Women | Men |
|---|---|---|
| **Baseline** | | |
| Ethnicity | | |
| Non-white | 134 (3.64) | 138 (3.74) |
| White | 3545 (96.36) | 3552 (96.26) |
| At least one parent had higher than O-level education | | |
| Yes | 3224 (55.29) | 3400 (54.76) |
| No | 2607 (44.71) | 2809 (45.24) |
| At least one parent part of lower social class (partly or unskilled occupation) | | |
| Yes | 1150 (23.76) | 1167 (22.87) |
| No (Both parents in professional, managerial or skilled occupations) | 3690 (76.24) | 3936 (77.13) |
| Mother married | | |
| Yes | 4807 (75.30) | 5100 (74.53) |
| No | 1577 (24.70) | 1743 (25.47) |
| Lived with both biological parents | | |
| Yes | 4489 (90.29) | 4830 (90.26) |
| No | 483 (9.71) | 521 (9.74) |
| **Early adulthood (ages 18–21)** | | |
| Longest relationship (at age 18) | | |
| More than 3 months | 1632 (72.18) | 1034 (58.78) |
| Less than or equal to 3 months | 629 (27.82) | 725 (42.22) |
| Living arrangements (at age 20) | | |
| One or both parents | 1200 (48.21) | 819 (51.51) |
| Partner and/or children | 307 (12.33) | 98 (6.16) |
| Other | 982 (39.45) | 673 (42.33) |
| Sexual preference (at age 21) | | |
| Asexual | 8 (0.37) | 6 (0.51) |
| Any same-sex preferences | 358 (16.63) | 160 (13.72) |
| 100% heterosexual | 1787 (83.00) | 1000 (85.76) |

sample. The scree plot plateaued between the second and third factor, and as the second factor had an eigenvalue close to 1 (0.847), we also attempted a two-factor solution with oblique rotation. This two-factor solution fit the data well, indicating plausible dimensions for (1) physical and psychological IPV and (2) sexual IPV. This suggests that analyses using a latent variable approach could reliably analyse these two factors. The factor analysis did not support a three-factor solution: the third factor had a low eigenvalue (0.182) and no items with a loading greater than 0.30. Overall, results were similar when factor analyses were run separately by gender (online supplementary appendix tables A3 and A4): all items loading highly onto a single factor and the same two-factor solution was identified for women and men.

As expected, the eight negative impacts were all positively correlated ($\rho$=0.297–0.893, online supplementary appendix table A5). These items were also negatively correlated with IPV having no impact, seeming funny or increasing perceptions of being loved, protected or wanted ($\rho$=−0.264 to −0.862). Finally, these three null or positive impacts were positively correlated ($\rho$=0.419–0.639). We, therefore, as planned, correlated the sum total of the negative impacts of IPV with the average frequency of IPV experiences among those who had experienced any IPV. As expected, experiencing more frequent IPV was strongly correlated with experiencing more negative impacts (n=1111): r=0.579, p<0.001.

### Overall prevalence

As shown in table 3, the most frequently experienced IPV was psychological (eg, 25% of participants reported humiliation, name-calling or shouting) and the least experienced was sexual (eg, 4% reported forced sex). Among those who experienced any IPV, the majority of violent acts (>78%) occurred after age 18 (see online supplementary appendix table A6 for more detail). Most participants reported at least one negative impact

**Table 3** Frequencies of 8 intimate partner violence (IPV) victimisation and impact items

| Victimisation items | Total N | N (%) | | | |
|---|---|---|---|---|---|
| | | **Never** | **Once** | **A few times** | **Often** |
| Told you who you could see and where you could go and/or regularly checked what you were doing and where you were (by phone or text) | 3268 | 2544 (77.85) | 124 (3.79) | 322 (12.91) | 178 (5.45) |
| Made fun of you, called you hurtful names, shouted at you | 3253 | 2422 (74.45) | 170 (5.23) | 530 (16.29) | 131 (4.03) |
| Used physical force such as pushing, slapping, hitting or holding you down | 3255 | 2768 (85.04) | 193 (5.93) | 235 (7.22) | 59 (1.81) |
| Used more severe physical force such as punching, strangling, beating you up, hitting you with an object | 3252 | 3075 (94.56) | 81 (2.49) | 68 (2.09) | 28 (0.86) |
| Pressured you into kissing/touching/something else | 3255 | 2981 (96.58) | 96 (2.95) | 146 (4.49) | 32 (0.98) |
| Physically forced you into kissing/touching/something else | 3250 | 3115 (95.85) | 68 (2.09) | 49 (1.51) | 18 (0.55) |
| Pressured you into having sexual intercourse | 3242 | 2876 (88.71) | 181 (5.58) | 152 (4.69) | 33 (1.02) |
| Physically forced you into having sexual intercourse | 3239 | 3118 (96.26) | 80 (2.47) | 32 (0.99) | 9 (0.28) |
| **Impact items** | **Total N** | **Never** | **Once** | **A few times** | **Often** |
| Scared or frightened in any way | 3221 | 2711 (84.17) | 191 (5.93) | 234 (7.26) | 85 (2.64) |
| **Impact items: only those who experienced at least 1 act of IPV** | **Total N** | **Yes** | | **No** | |
| Upset/unhappy | 1148 | 900 (78.40) | | 248 (21.60) | |
| Angry/annoyed | 1139 | 857 (75.24) | | 282 (24.76) | |
| Made me feel sad | 1142 | 813 (71.19) | | 329 (28.81) | |
| Affected my work/studies | 1141 | 799 (70.03) | | 342 (29.97) | |
| Anxious | 1133 | 495 (43.69) | | 638 (56.31) | |
| Depressed | 1138 | 418 (36.73) | | 720 (63.27) | |
| No effect/not bothered | 1133 | 206 (18.18) | | 927 (81.82) | |
| Made me drink more alcohol/take more drugs | 1138 | 168 (14.76) | | 970 (85.24) | |
| Thought it was funny | 1132 | 158 (13.96) | | 974 (86.04) | |
| Felt loved/protected/wanted | 1135 | 148 (13.04) | | 987 (86.96) | |

following IPV, with the most common being feeling upset (78%) or angry (75%). The least common impacts of IPV were the positive ones: 13% of participants reported that the violence made them feel loved, protected or wanted; 14% found the violence amusing.

Overall, 37% of participants reported experiencing any IPV and 29% experienced any IPV after age 18. The mean number of IPV acts experienced among those who experienced any violence, ranging from 1 to 8, was 3.004 (SD=2.108) overall and 2.167 (SD=1.644) after age 18. The mean number of negative impacts, ranging from 0 to 8, was 3.950 (SD=2.371) among those who had experienced any IPV and 2.944 (SD=2.633) among those who had experienced IPV after age 18.

### Gender differences

As shown in table 4, for all IPV victimisation items, regardless of whether lifetime or early adulthood (ages 18–21) was considered, significantly more women experienced violence than men. The largest percentage difference was for the lifetime prevalence of coerced sex (15% women, 4% men). Moreover, significantly more women than men reported experiencing all negative impacts of IPV, apart from substance use where there was no difference. The greatest percentage difference was in feeling scared because of their partner (56% women, 14% men in their lifetime). In contrast, more men than women reported that the IPV they experienced was funny or had no effect on them. Finally, every test indicated that women experienced more frequent and severe IPV overall than men, in both their lifetimes and early adulthood (table 5): women experienced more frequent and a greater number of acts of IPV compared to men; more women than men experienced any IPV (with or without negative impact); and, among those who had experienced any IPV, women experienced more negative impacts than men.

**Table 4** Gender differences in IPV victimisation and impact items

| Victimisation items | Lifetime | | | Ages 18–21 | | |
| --- | --- | --- | --- | --- | --- | --- |
| | Women (n=2050) | Men (n=1108) | $\chi^2$ (p value) | Women (n=2014) | Men (n=1092) | $\chi^2$ (p value) |
| Told you who you could see, where you could go or regularly checked what you were doing and where you were | 510 (24.88) | 196 (17.69) | 21.41 (<0.001) | 346 (17.18) | 152 (13.92) | 5.59 (.018) |
| Made fun of you, called you hurtful names, shouted at you | 596 (29.07) | 210 (18.95) | 38.75 (<0.001) | 443 (22.00) | 166 (15.20) | 20.74 (<0.001) |
| Used physical force such as pushing, slapping, hitting or holding you down | 362 (17.66) | 106 (9.57) | 37.31 (<0.001) | 245 (12.16) | 82 (7.51) | 16.29 (<0.001) |
| Used more severe physical force such as punching, strangling, beating you up, hitting you with an object | 142 (6.93) | 31 (2.80) | 23.68 (<0.001) | 96 (4.77) | 22 (2.01) | 14.67 (<0.001) |
| Pressured you into kissing/touching/something else | 240 (11.71) | 26 (2.35) | 81.70 (<0.001) | 144 (7.15) | 20 (1.83) | 40.05 (<0.001) |
| Physically forced you into kissing/touching/something else | 125 (6.10) | 7 (0.63) | 53.65 (<0.001) | 72 (3.57) | 5 (0.46) | 28.46 (<0.001) |
| Pressured you into having sexual intercourse | 313 (15.27) | 43 (3.88) | 93.25 (<0.001) | 192 (9.53) | 36 (3.30) | 40.49 (<0.001) |
| Physically forced you into having sexual intercourse | 114 (5.56) | 5 (0.45) | 51.79 (<0.001) | 64 (3.18) | 5 (0.46) | 24.12 (<0.001) |

| Impact items (among those who experienced any IPV) | Women (n=800) | Men (n=292) | $\chi^2$ (p value) | Women (n=552) | Men (n=221) | $\chi^2$ (p value) |
| --- | --- | --- | --- | --- | --- | --- |
| Scared | 444 (55.50) | 40 (13.70) | 151.47 (<0.001) | 279 (50.54) | 31 (14.03) | 87.61 (<0.001) |
| Upset/unhappy | 684 (85.50) | 179 (61.30) | 75.58 (<0.001) | 465 (84.24) | 141 (63.80) | 38.92 (<0.001) |
| Angry/annoyed | 625 (78.12) | 195 (66.78) | 14.72 (<0.001) | 441 (79.89) | 152 (68.78) | 10.91 (0.001) |
| Made me feel sad | 621 (77.62) | 157 (53.77) | 59.44 (<0.001) | 425 (76.99) | 122 (55.20) | 36.22 (<0.001) |
| Affected my work/studies | 275 (34.38) | 51 (17.47) | 29.21 (<0.001) | 187 (33.88) | 38 (17.19) | 21.28 (<0.001) |
| Anxious | 406 (50.75) | 73 (25.00) | 57.60 (<0.001) | 272 (49.28) | 60 (27.15) | 31.53 (<0.001) |
| Depressed | 329 (41.12) | 69 (23.63) | 28.27 (<0.001) | 231 (41.85) | 52 (23.53) | 22.82 (<0.001) |
| No effect/not bothered | 109 (13.63) | 89 (30.48) | 40.94 (<0.001) | 74 (13.41) | 59 (26.70) | 19.57 (<0.001) |
| Made me drink more alcohol/take more drugs | 127 (15.88) | 34 (11.64) | 3.05 (0.081) | 92 (16.67) | 30 (13.57) | 1.14 (0.287) |
| Thought it was funny | 64 (8.00) | 92 (31.51) | 96.53 (<0.001) | 48 (8.70) | 67 (30.32) | 58.26 (<0.001) |
| Felt loved/protected/wanted | 101 (12.62) | 41 (14.04) | 0.38 (0.538) | 80 (14.49) | 32 (14.48) | 0.00 (0.996) |

Victimisation items were coded as 1=experienced at least once, 0=never experienced. Impact items were 1=yes, 0=no.
IPV, intimate partner violence.

**Table 5** Summary statistics for comparisons between women and men on overall IPV victimisation and impact

| Item | Lifetime | | | | | | Ages 18–21 | | | | | |
| | Women | | Men | | | | Women | | Men | | | |
| | N | M (SD) or N (%) | N | M (SD) or N (%) | t(df) or $\chi^2$ | P value | N | M (SD) or N (%) | N | M (SD) or N (%) | t(df) or $\chi^2$ | P value |
|---|---|---|---|---|---|---|---|---|---|---|---|---|
| Mean frequency of IPV experiences (SD) | 2128 | 0.28 (0.50) | 1145 | 0.12 (0.25) | 12.61 (3252.18) | <0.001 | 2128 | 0.19 (0.39) | 1145 | 0.10 (0.24) | 7.58 (3219.11) | <0.001 |
| Mean no of IPV acts experienced (SD) | 2024 | 1.41 (2.19) | 1096 | 0.60 (1.22) | 13.16 (3115.22) | <0.001 | 2014 | 0.75 (1.47) | 1092 | 0.42 (0.97) | 7.55 (2996.44) | <0.001 |
| Any IPV (N, %) | 2128 | 884 (41.54) | 1145 | 330 (28.82) | 51.62 | <0.001 | 2128 | 683 (32.10) | 1145 | 275 (24.02) | 23.47 | <0.001 |
| Any IPV with a negative impact (N, %) | 2126 | 788 (37.06) | 1144 | 236 (20.63) | 93.41 | <0.001 | 2126 | 608 (28.60) | 1145 | 200 (17.47) | 49.57 | <0.001 |
| Mean no of negative impacts of IPV (SD) | 800 | 4.39 (2.27) | 292 | 2.73 (2.21) | 10.75 (1090) | <0.001 | 746 | 3.21 (2.72) | 279 | 2.24 (2.24) | 5.77 (602.84) | <0.001 |

All t-tests were two-group t-tests with unequal variances, apart from 'number of negative impacts of IPV' for the overall sample, which did not have unequal variances between men and women (ie, Levene's test was statistically non-significant).
IPV, intimate partner violence.

## DISCUSSION

This study estimated the prevalence of physical, psychological and sexual IPV in a UK birth cohort during early adulthood using a novel measure. The prevalence of IPV was high: 37% of participants had experienced any IPV in their lifetime and 29% had experienced IPV between ages 18 and 21. As in previous research, the most commonly experienced violence was psychological and the least commonly experienced was sexual.[5] Over three-quarters of those who had experienced IPV had experienced this violence when they were aged 18 or older. This aligns with the broader IPV literature, which has found that early adulthood is an especially high-risk period for experiencing IPV.[19] Most participants who had experienced IPV reported more than one negative psychological impact, with the most common being feeling upset or angry. The least common outcomes of IPV were finding the violence amusing or feeling more loved, wanted or protected.

We found strong and consistent gender differences: for all types of violent behaviours, women experienced more frequent IPV than men, both in their lifetime and early adulthood. As in other prevalence surveys, the most dramatic differences between women and men were on sexual violence items.[5] For instance, the proportion of women who had ever experienced coerced sex was more than four times that of men. Moreover, significantly more women than men reported experiencing negative psychosocial impacts from IPV. For example, the proportion of women who felt afraid of their partner was more than four times that of men. Similar proportions of women and men reported that their alcohol and substance use increased after experiencing IPV. The evidence on whether there are gender differences in substance use following IPV is inconsistent[27]; one possible explanation for similar proportions is the greater psychological impacts of IPV among women balance with the greater baseline tendency among men to use substances.

In contrast, the proportion of men who found their experiences of IPV amusing was more than three times that of women. More than double the proportion of men also reported that this violence did not affect them. Together with the gender differences in the negative impacts of IPV, this suggests that women experience more severe IPV than men, which is more difficult to trivialise and more likely to cause psychological harm. This extends a large body of evidence demonstrating that women experience the majority of severe consequences of IPV and are more likely to have controlling, violent partners.[2 28] At the same time, participants' reporting on the impacts and interpretations of their IPV experiences may have been influenced by gender role socialisation. Women may have more readily reported the negative consequences of violence from their partners whereas men may have felt more pressure to minimise their experiences and deny negative consequences due to internalised concepts of masculinity, for instance, as strong and powerful and femininity as vulnerable and weak.[29–31]

Future survey research could explore this hypothesis by including questions on traditional gender role attitudes.

Our findings contribute to the broader debate on gender asymmetry in IPV. Studies using the Conflict Tactics Scale or family conflict surveys tend to find equivalent prevalence estimates among women and men[32]; whereas crime or clinical surveys tend to find that women experience more IPV than men.[11 32] Our data came from a community-based birth-cohort study, using a measure without reference to crime or conflict resolution, which minimises these priming biases. Our findings demonstrate that women experience more frequent and severe IPV than men, but also confirm that a considerable number of men experience violence from their partners. Therefore, our results support the continued research and advocacy enterprise for IPV against women in particular, while also demonstrating the need for resources to continue to be developed for both women and men.

The new measure for IPV tested in this study showed excellent internal consistency and a strong, positive correlation with negative impacts of IPV, indicating convergent validity. The exploratory factor analysis suggested that the measure could be reliably analysed as a single dimension of IPV or as two—(1) physical or psychological IPV and (2) sexual IPV. As the scale items all loaded highly onto a single factor, analysing a factor-based score would be appropriate and have the benefit of maintaining the measure's original scaling for more intuitive interpretation.[33] Overall, factor structures were equivalent among women and men. This should be confirmed in new samples, including tests of gender invariance, which overall has been understudied in the literature.[34]

### Strengths and limitations

Study limitations include, first, not measuring the age of first occurrence of IPV before age 18. Second, the definition of intimate partner used was broad, from casual sex partners to long-term relationships. Since this could capture sexual violence by an acquaintance, future research should use a more constrained definition (eg, dating or marital partner). Third, the ALSPAC instrument did not measure specific instances of IPV or specific relationships; it is, therefore, unclear whether IPV was experienced by multiple perpetrators or repeatedly during a single relationship. We are also unable to determine which types or instances of IPV caused the impacts reported. Although more time intensive, it would be useful if future uses of the ALSPAC instrument allowed participants to indicate the perpetrator(s) and impact(s) of each experience of IPV. Obtaining more detailed information on IPV events and the relationship context would help determine intent and precipitants to inform directions for intervention research. Relatedly, data from participants' partners or an equivalent measure of IPV perpetration were not collected in ALSPAC. However, in the absence of sampling partners, self-reported victimisation is a more sensitive measure of IPV than self-reported perpetration.[35 36] Moreover, although IPV experiences may have involved the use of violence as well, that a greater proportion of women experienced nearly every measured negative impact from IPV compared with men suggests important differences in the severity and experience of IPV that remain critical to consider both for research and clinical practice.

Fourth, the IPV impacts measured were mainly psychological, to the exclusion of further physical (eg, injury) or socioeconomic consequences (apart from work or studies being affected). Additionally, the impact items were largely one-word terms such as depressed and anxious, which are more vulnerable to social desirability bias and internalised gender concepts as opposed to a scale of items measuring depression or anxiety.[37] Nevertheless, in longer surveys such as those used in ALSPAC, it may not be feasible to include more exhaustive measures of IPV impacts. Fifth, ALSPAC did not include alternative IPV measures to further evaluate the measure's convergent validity. Assessing convergence with long-form IPV measures, in particular, may be useful to determine if scale length or breadth has any impact on sensitivity or gender differences. Finally, higher socioeconomic positions and white persons are over-represented in this sample: the generalisability of our results to the greater UK population or other contexts requires further investigation.

Despite these limitations, our study has a number of strengths. The IPV measure used was brief and could therefore be implemented in surveys measuring a rich set of potential IPV predictors. Both women and men were sampled, allowing for analyses of gender differences. Although age of first occurrence was not measured, participants indicated whether IPV occurred before and/or after age 18, which made it possible to compute a current prevalence of IPV (last 3 years) and allows for analyses of temporal relationships between antecedents and early adulthood IPV. Finally, our analyses were thorough and demonstrated consistent results, allowing for firm conclusions on the reliability of the IPV measure and gender differences in IPV in the ALSPAC cohort.

In conclusion, we found that more than one-third of participants engaged in a cohort study for over 20 years in the UK had experienced IPV by early adulthood. Women consistently experienced more frequent and severe IPV than men by all measures, suggesting important gender differences in the burden of this violence. The ALSPAC measure for IPV victimisation showed strong indicators of reliability and validity, demonstrating its appropriateness for further aetiological studies.

**Acknowledgements** The authors would like to thank all the families who took part in ALSPAC, the midwives who helped recruit them and the whole ALSPAC team, including the interviewers, computer and laboratory technicians, clerical workers, research scientists, volunteers, managers, receptionists and nurses.

**Contributors** ARY and DKH designed this study. ARY conducted the analyses, interpreted the data and drafted the manuscript. GF, JH, AF and DKH provided comments on the manuscript. All authors have read and approved this final version. This manuscript is the work of the authors, who will serve as guarantors for its contents.

**Funding** The UK Medical Research Council and Wellcome (Grant ref: 102215/2/13/2) and the University of Bristol provide core support for ALSPAC. A comprehensive list of grants funding is available on the ALSPAC website (http://www.bristol.ac.uk/alspac/external/documents/grant-acknowledgements.pdf); this research was specifically funded by NHS Bristol Clinical Commissioning Group (RP-PG-0108-10048). ARY is funded by The Rhodes Trust and the Canadian Institutes of Health Research (DFS152265). Data access costs were additionally supported by the Department of Social Policy and Intervention Student Support Fund. AF and JH work at the MRC Integrative Epidemiology Unit which receives infrastructure funding from the UK Medical Research Council (MRC) (MC_UU_12013). AF is funded by a UK MRC fellowship awarded to AF (MR/M009351/1). GF is supported by the NIHR Biomedical Research Centre at University Hospitals Bristol NHS Foundation Trust and the University of Bristol.

**Disclaimer** The views expressed in this manuscript are those of the author(s) and not necessarily those of the NHS, the National Institute for Health Research or the Department of Health.

**Competing interests** None declared.

**Patient consent for publication** Not required.

**Ethics approval** Ethical approval was obtained from the ALSPAC Ethics and Law Committee and local research ethics committees.

**Provenance and peer review** Not commissioned; externally peer reviewed.

**Data sharing statement** Data underlying the results of this study are available from ALSPAC, based at the University of Bristol. Requests for data should be submitted to the ALSPAC Executive Committee at https://proposals.epi.bristol.ac.uk/orsenttoalspac-data@bristol.ac.uk/. Details on all data are available through a fully searchable data dictionary at http://www.bristol.ac.uk/alspac/researchers/our-data/. Analysis syntax is stored by ARY and available on request.

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
