## [Reviewer comments · BMJ Open]

ARTICLE DETAILS

TITLE (PROVISIONAL)	Intimate partner violence victimisation in early adulthood: psychometric properties of a new measure and gender differences in the Avon Longitudinal Study of Parents and Children
AUTHORS	Yakubovich, Alexa; Heron, Jon; Feder, Gene; Fraser, Abigail; Humphreys, David

VERSION 1 - REVIEW

REVIEWER	Laura E. Watkins Emory University, United States
REVIEW RETURNED	27-Sep-2018

GENERAL COMMENTS	The manuscript entitled "Intimate partner violence victimization in early adulthood: psychometric properties of a new measure and gender differences in the Avon Longitudinal Study of Parents and Children" by Yakubovich et al. examines a new measure of intimate partner violence among a sample of young adults in the United Kingdom. Strengths of this manuscript include its examination of an important topic and its large sample. However, the manuscript includes several areas of concern, in addition to areas for clarification, which are described below. 1. My first concern is related to the analyses. It appears that the authors had a good inclination of what the factor structure of the measure would be before analyzing the data, which suggests confirmatory factor analysis (CFA) would be more appropriate than exploratory factor analysis. Relatedly, this manuscript would be improved if the authors tested for gender invariance within a CFA to determine if the measure is assessing the same construct across genders.2. Many measures of intimate partner violence (IPV) have already been developed. There are also several short or screener measures that are not mentioned in the introduction, such as the Brief CTS and the E-HITS. The measure used in the current study is not very different from these brief measures. Please provide a stronger rationale for the development of this new measure in a field where many measures of IPV already exist. Also, please provide rationale for half of the questions being about sexual violence instead of an equal number of questions for each type of violence (psychological, physical, and sexual).3. It does not appear that IPV perpetration was assessed. Research demonstrates that much IPV, especially in nonclinical samples, such as the one used in this study, is bidirectional. This seems like important information to collect. Information on perpetration would help clarify if women were greater reporters of
---

	IPV overall (reporting more victimization and perpetration) or if they only reported greater victimization. 4. The evidence for the measure's validity is meager. Are there any other measures in this study that could be used for convergent or divergent validity? It seems rather strong to state that this study shows validity of this measure when it was only correlated with the impact items. 5. The response options for the IPV measure are unclear. Please clarify in the description of the measure. Relatedly, please clarify what "average frequency of IPV experiences (maximum=3)" (pg 6) means and how the mean number of IPV acts, which has a maximum of 8, was calculated. 6. In the discussion, the authors note that the CTS tends to find equivalent prevalence estimates and cites differences between samples. One difference with the CTS worth noting is that individuals report on their own and their partners' behaviors rather than solely on victimization. The CTS also includes many more items than the current study, which also could be a reason for potential gender differences.
--	--

REVIEWER	Andrea Warner Stidham Assistant Professor, College of Nursing Kent State University, Kent, OHIO, USA
REVIEW RETURNED	03-Oct-2018

GENERAL COMMENTS	Very well written manuscript on a timely and important topic. Excellent description of aims, purpose, methods, results, and discussion. A potential limitation that is not addressed is the representativeness of the sample to the broader U.K. population. While the findings suggest the tool to be useful in a White, non-impooverished sample, would it offer similar results in other populations? While I do not believe this is a "fatal flaw", the implications of such a homogenous sample could be considered and/or addressed in the manuscript. Also, two areas of consideration: 1) pg. 12, beginning line 37--sentence is unclear and would be better reworded and 2) pg. 16, line 12 (the note refers to Table X--this needs to be updated to reflect the referred table).
---

VERSION 1 – AUTHOR RESPONSE

Reviewer #1:

1. The manuscript entitled "Intimate partner violence victimization in early adulthood: psychometric properties of a new measure and gender differences in the Avon Longitudinal Study of Parents and Children" by Yakubovich et al. examines a new measure of intimate partner violence among a sample of young adults in the United Kingdom. Strengths of this manuscript include its examination of an important topic and its large sample. However, the manuscript includes several areas of concern, in addition to areas for clarification, which are described below.

My first concern is related to the analyses. It appears that the authors had a good inclination of what the factor structure of the measure would be before analyzing the data, which suggests confirmatory factor analysis (CFA) would be more appropriate than exploratory factor analysis. Relatedly, this manuscript would be improved if the authors tested for gender invariance within a CFA to determine if the measure is assessing the same construct across genders.

Thank you very much to the reviewer for raising this point. We agree that gender invariance is an important and often neglected component of IPV measurement. However, given that this study represents the first time that this novel measure of IPV was implemented and evaluated, we believe that exploratory factor analysis, rather than confirmatory factor analysis, was the most appropriate method according to standard practice.¹ Any of a one-, two-, or three-factor solution would have been plausible with this measure and we did not have hypotheses regarding which solution would be the best fit. Moreover, it was also possible that the items would not hang well together or that different types of violence would group together (e.g., psychological and sexual rather than psychological and physical). The exploratory nature of these analyses is described in our method section, which we have now revised to clarify that the dimensionality of the scale was not clear nor previously tested prior to analysis (page 5):

As the scale's dimensionality was unknown, we conducted an exploratory factor analysis using this matrix.²⁶ We decided the appropriate number of factors based on their eigenvalues (using Kaiser's criterion that >1 indicates a viable factor), scree plot, and theoretical plausibility.²⁷ If a two (or more) factor solution was favourable, we decided a priori to use oblique rotation since we expected differing dimensions of abuse to correlate.

Indeed, although we found that a one-factor solution was the best fit with the data, a two-factor solution was also viable, as discussed in both our results (page 6, paragraph 2) and discussion sections (page 12, paragraph 1).

As conducting a confirmatory factor analysis in the same sample used for exploratory factor analysis would not be advisable,¹ to address the reviewer's valid comments regarding possible gender differences in IPV measurement, we ran the exploratory factor analysis separately for women and men. This is now reflected in our method section (page 5):

To test for possible gender differences in factor solutions, we also ran the exploratory factor analysis separately for women and men.

Overall, the factor solutions and loadings were similar by gender, as is now described in our results section (page 6):

Overall, results were similar when factor analyses were run separately by gender (Tables A3A4): all items loading highly onto a single factor and the same two-factor solution was identified for women and men.

The following tables, now added to the appendix, show the factor solutions and overall patterns of factor loadings for women and men separately:

Table A3: Exploratory factor analysis (women only)

Item	Single factor solution		Two-factor solution (r=63.86%)		
	Factor 1	Uniqueness	Factor 1	Factor 2	Uniqueness
Control	.786	.382	-	.714	.322
Humiliate	.805	.351	-	.863	.220
Push, slap	.886	.215	-	.887	.096
Punch, strangle	.860	.260	-	.928	.106
Coerced touch	.858	.264	.948	-	.107

Forced touch	.878	.230	.929	-	.093
Coerced sex	.860	.261	.853	-	.162
Forced sex	.892	.205	.755	-	.158
Factor eigenvalue:	5.832		5.832	0.903	
Proportion of variance explained	.853		.853	.132	

Note. N=2,050 women. Method is principal factors using a polychoric correlation matrix. Two-factor solution uses obliquemax). Factor loadings <.4 are suppressed. Full rotation (pro item descriptions are shown in Table 1.

Table A4: Exploratory factor analysis (men only)

Item	Single factor solution		Two-factor solution (r=59.60%)		
	Factor 1	Uniqueness	Factor 1	Factor 2	Uniqueness
Control	.790	.377	-	.751	.294
Humiliate	.749	.439	-	.896	.248
Push, slap	.869	.245	-	.739	.188
Punch, strangle	.776	.398	-	.934	.189
Coerced touch	.876	.233	.928	-	.086
Forced touch	.926	.142	.646	-	.132
Coerced sex	.822	.325	.876	-	.192
Forced sex	.822	.325	.992	-	.104
Factor eigenvalue:	5.518		5.518	1.049	
Proportion of variance explained	.690		.690	.131	

Note. N=1,108 men. Method is principal factors using a polychoric correlation matrix, forced to be positive definite. Two-factor solution uses oblique rotation (promax). Factor loadings <.4 are suppressed. Full item descriptions are shown in Table 1.

To discuss these findings and further address the reviewer's important comments regarding the utility of testing for gender invariance in future investigations, we have also added the following to our discussion section (page 12):

Overall, factor structures were equivalent among women and men. This should be confirmed in new samples, including tests of gender invariance, which overall has been understudied in the literature.³⁵

- Many measures of intimate partner violence (IPV) have already been developed. There are also several short or screener measures that are not mentioned in the introduction, such as the Brief CTS and the E-HITS. The measure used in the current study is not very different from these brief measures. Please provide a stronger rationale for the development of this new measure in a field where many measures of IPV already exist. Also, please provide rationale for half of the questions being about sexual violence instead of an equal number of questions for each type of violence (psychological, physical, and sexual).

We agree this is a critical point. We have added a citation to the short-form of the Conflict Tactics Scale, however, as we address in our introduction, this scale has been heavily critiqued for measuring intimate partner violence only within the context of conflicts and failing to measure the impacts of this violence (page 3, paragraph 2):

The most commonly used measure is currently the Conflict Tactics Scale,⁸⁻¹⁰ which measures specific behaviours by a current or previous dating, cohabiting, or marital partner. However, the Conflict Tactics Scale has been criticised for measuring IPV only within the context of conflicts or disagreements and not measuring the intent (e.g., self-defense or harm) or impact of violence.^{11 12}

We have also added a citation to the E-HITS screening tool, however, as with many other available IPV measures (both short- and long-form) this instrument does not measure important

forms of psychological violence (in this case, controlling behaviour). The ALSPAC measure uniquely captures the prevalence of psychological (both emotional abuse and controlling behaviour), sexual (coerced or forced sexual activity), and physical IPV (pushing, hitting, slapping, and more severe forms of violence such as punching, strangling, beating up, or use of objects) as well as the impacts of this violence in a sample of both women and men. The following revised section of our introduction clarifies this point (page 3) and the novelty of the current study:

Other validated scales include the Composite Abuse Scale,¹³ WHO multi-country survey,¹⁴ Abusive Behavior Inventory,¹⁵ Severity of Violence Against Women Scale,¹⁶ Measure of Wife Abuse,¹⁷ and the Extended-Hurt/Insult/Threaten/Scream tool.¹⁸ However, several of these do not measure psychological IPV (including controlling behaviour)^{14 16 18} and most are relatively long (>30 items),^{13 15-17} risking response burden in larger or repeated-measures surveys.

In recent years, in response to the criticisms and limitations of existing measures, short-form measures of physical, psychological, and sexual IPV have been developed with emerging evidence of validity (e.g., among Canadian women: the short-form Composite Abuse Scale¹⁹). This study is the first psychometric evaluation of a short-form measure for physical, psychological, and sexual IPV developed in the UK, which uniquely also collected data on the impacts of this violence and sampled both women and men.

Regarding the items in the IPV measure, the rationale is the distinction between being pressured and being forced into sexual activity. In a sense, being pressured to have sexual contact or sexual intercourse is a type of psychological or emotional coercion and distinct from physical sexual assault. The sexual pressure/force questions, as well as the emotional abuse/control items, used in the ALSAPC measure were adapted from the NSPCC/University of Bristol Young People's Relationships questionnaire (Barter et al 2009). These items were developed with a young person's advisory group (YPAG), consisting of 12 young people, who advised on the content of the questionnaire, including the ratio of physical/emotional/sexual abuse questions. Additionally, the final IPV measure implemented in ALSPAC was piloted with the ALSPAC participant advisory group. To clarify the overlap in the types of IPV measured by the items, we have revised 'type of IPV' for each of the sexual pressure items to 'sexual/psychological' in Table 1 (page 4):

Table 1: IPV victimisation and impact items

Order	Victimisation items: How often altogether have any of your partners ever done any of the following to you and how old were you?	Type of IPV
1	Told you who you could see and where you could go and/or regularly checked what you were doing and where you were (by phone or text)?	Psychological
2	Made fun of you, called your hurtful names, shouted at you?	Psychological
3	Used physical force such as pushing, slapping, hitting or holding you down?	Physical
4	Used more severe physical force such as punching, strangling, beating you up, hitting you with an object?	Physical
5	Pressured you into kissing/touching/something else?	Sexual/ psychological
6	Physically forced you into kissing/touching/something else?	Sexual
Order	Victimisation items: How often altogether have any of your partners ever done any of the following to you and how old were you?	Type of IPV
7	Pressured you into having sexual intercourse?	Sexual/ psychological
8	Physically forced you into having sexual intercourse?	Sexual

3. It does not appear that IPV perpetration was assessed. Research demonstrates that much IPV, especially in nonclinical samples, such as the one used in this study, is bidirectional. This seems like important information to collect. Information on perpetration would help clarify if

women were greater reporters of IPV overall (reporting more victimization and perpetration) or if they only reported greater victimization.

A measure of IPV perpetration equivalent in rigour and scope to the IPV victimisation measure was not implemented within ALSPAC. We acknowledge that a limitation of this study is the lack of data on the perpetrator(s) of the IPV experienced by participants and the specific instances of this violence, including intent and precipitants. However, there are several reasons it is of value to focus on measuring and evaluating self-reports of experiencing IPV (i.e., victimisation), both in the current study and beyond. First, as demonstrated in Archer and colleagues' meta-analysis of self- and partner-reports as well as later studies, people tend to self-report victimisation more than they self-report perpetration, making the former a more sensitive measure for IPV. This is especially important in the context of studies that are unable to sample both members of the couple, which in long-term cohort studies is often unfeasible. Second, we believe that understanding the context of IPV is critical to interpreting the significance of (bi)directionality. The fact that a greater proportion of women experienced every measured negative impact from IPV as compared to men suggests important differences in the severity and experience of IPV that are critical to consider both for research and clinical practice. In order to more effectively articulate these points and address the reviewer's concerns regarding potential bidirectionality, we have revised our discussion of study limitations relevant to this comment, which now reads as follows (page 12):

Third, the ALSPAC instrument did not measure specific instances of IPV or specific relationships; it is therefore unclear whether IPV was experienced by multiple perpetrators or repeatedly during a single relationship. We are also unable to determine which types or instances of IPV caused the impacts reported. Although more time intensive, it would be useful if future uses of the ALSPAC instrument allowed participants to indicate the perpetrator(s) and impact(s) of each experience of IPV. Obtaining more detailed information on IPV events and the relationship context would help determine intent and precipitants to inform directions for intervention research. Relatedly, data from participants' partners or an equivalent measure of IPV perpetration were not collected in ALSPAC. However, in the absence of sampling partners, self-reported victimisation is a more sensitive measure of IPV than self-reported perpetration.^{36,37} Moreover, although IPV experiences may have involved the use of violence as well, that a greater proportion of women experienced nearly every measured negative impact from IPV compared to men suggests important differences in the severity and experience of IPV that remain critical to consider both for research and clinical practice.

4. The evidence for the measure's validity is meager. Are there any other measures in this study that could be used for convergent or divergent validity? It seems rather strong to state that this study shows validity of this measure when it was only correlated with the impact items.

Previous validation studies of IPV measures have used mental health measures to test convergent validity. Although these are available in ALSPAC, we believe testing the correlation between general mental health (e.g., depressive symptomatology) and IPV is a valid research question in itself as opposed to an appropriate or more reliable indicator of convergent validity. For this reason, we instead rely on the correlation between the reported frequency of experiencing IPV based on the ALSPAC measure and the number of reported negative impacts experienced from IPV to evaluate convergent validity (as described on page 5, paragraph 1). However, it would be valuable for future research to further test the convergent validity of the ALSPAC IPV measure with other measures of IPV not available in the study, which we now address in our discussion section as follows (page 12):

Fifth, ALSPAC did not include alternative IPV measures to further evaluate the measure's convergent validity. Assessing convergence with long-form IPV measures in particular may be useful to determine if scale length or breadth has any impact on sensitivity or gender differences.

5. The response options for the IPV measure are unclear. Please clarify in the description of the measure. Relatedly, please clarify what "average frequency of IPV experiences (maximum=3)" (pg 6) means and how the mean number of IPV acts, which has a maximum of 8, was calculated.

Thank you pointing this out. Although the response options are included in the legend for Table 1, we have now also clarified these in text as follows (page 4):

Participants indicated the frequency of each item (coded 0=never to 3=often) and whether this occurred before and/or after age 18, allowing for measurement of temporality.

We have also now added the following clarification as to how we computed the summary IPV scores (page 5):

For aim two, we computed the prevalence and frequency of IPV experiences and impacts. This involved computing the mean of participants' scores across the eight IPV items (reflecting the average frequency of IPV experiences, 0-3), the mean number of IPV acts experienced at least once (0-8), the mean number of negative impacts from IPV experienced (0-8), the proportion of participants who experienced any IPV, and the prevalence of any IPV with at least one negative impact.

6. In the discussion, the authors note that the CTS tends to find equivalent prevalence estimates and cites differences between samples. One difference with the CTS worth noting is that individuals report on their own and their partners' behaviors rather than solely on victimization. The CTS also includes many more items than the current study, which also could be a reason for potential gender differences.

Thank you for highlighting these additional factors that may underlie differences between studies using the CTS and other measures. However, we would like to note that many studies using the CTS only use self-reports of victimisation and many long form scales in crime or clinical surveys have found gender asymmetry in IPV (e.g., the NISVS). However, to ensure that we have addressed these concerns, we have added to our limitation section that we do not have equivalent data on participants' partners or perpetration, as discussed in response to comment 3. In addition, we now also discuss that participants may respond differently to shortform measures as compared to long-form measures, as shown in response to comment 4.

Reviewer #2:

1. Very well written manuscript on a timely and important topic. Excellent description of aims, purpose, methods, results, and discussion. A potential limitation that is not addressed is the representativeness of the sample to the broader U.K. population. While the findings suggest the tool to be useful in a White, non-impooverished sample, would it offer similar results in other populations? While I do not believe this is a "fatal flaw", the implications of such a homogenous sample could be considered and/or addressed in the manuscript.

Thank you very much for these comments. We have now addressed limitations on the generalisability of our study in the discussion as follows (page 12):

Finally, higher socioeconomic positions and White persons are over-represented in this sample: the generalisability of our results to the greater UK population or other contexts requires further investigation.

We have also updated the summary of the strengths and limitations of our study to include a note on generalisability (page 2):

Details on specific incidents or perpetrators of intimate partner violence were not measured and the generalisability of study findings to the national population and other contexts should be investigated.

2. Also, two areas of consideration: 1) pg. 12, beginning line 37--sentence is unclear and would be better reworded

We have now revised the sentence for clarity (page 12):

Nevertheless, in longer surveys such as those used in ALSPAC, it may not be feasible to include more exhaustive measures of IPV impacts.

3. and 2) pg. 16, line 12 (the note refers to Table X--this needs to be updated to reflect the referred table).

Thank you, we have now corrected this note for Table A1 in the appendix:

Table A1: Polychoric correlations between ordinal IPV items

Note. N=3,158. Response categories for all variables were 0=never, 1=once, 2=a few times, and 3=often. Full item descriptions are shown in Table 1.

VERSION 2 – REVIEW

REVIEWER	Laura E. Watkins Emory University School of Medicine, U.S.
REVIEW RETURNED	07-Dec-2018
GENERAL COMMENTS	The authors have adequately addressed all my previous comments.